# Exploring Efficient and Simple Initialization Strategies for Bayesian Optimization with SETUP-BO

**Seyed Ali YaghoubNejad**
Department of Computer Engineering
Sharif University of Technology
`yaghoubnejad@sharif.edu`

**Mohammad Taghi Manzuri**
Department of Computer Engineering
Sharif University of Technology
`manzuri@sharif.edu`

## Abstract

This paper studies the effectiveness of random and grid initialization strategies in SETUP-BO, a self-tuning Bayesian optimization algorithm. Our experiments on benchmark functions compare the performance of these initialization strategies to deterministic initialization. The results show that random initialization outperforms other methods, indicating that it can enhance the performance of BO.

## 1 Introduction

Optimization methods can be classified into different categories, such as first-order, higher-order, and derivative-free, based on the gradient information they use (Sun et al., 2019). *Bayesian optimization* (BO) (Mockus & Mockus, 1991) is a powerful framework for optimizing unknown or expensive-to-evaluate functions. In this paper, we investigate the effectiveness of two alternative initialization strategies for SETUP-BO[1], a recently proposed self-tuning BO algorithm (Vasconcelos et al., 2022). Specifically, we compare the deterministic initialization strategy to the random initialization (Bergstra & Bengio, 2012) and grid initialization strategies, hypothesizing that these simple strategies can outperform the original for certain types of functions.

In order to evaluate the efficacy of these initialization strategies, we have conducted several experiments on two widely-used benchmark functions in the field of optimization research. Our findings demonstrate that, for these specific benchmarks, the random initialization strategy consistently outperforms alternative methods. This highlights the potential for straightforward strategies to significantly enhance the performance of BO. These results are presented in Section 3.

## 2 Methodology

BO relies on the *surrogate model*, usually a Gaussian process, to capture the characteristics of the objective function (see A.1) and the *acquisition function* to suggest the next candidate for assessment (see A.2). To enhance the performance of BO, Auer et al. (1995) introduced *GP-Hedge* (see A.3) algorithm that uses a portfolio of acquisition functions and associates a score to each function based on its performance since the initial step.

However, GP-Hedge has a limitation; it remembers the initial poor performance of each function. Accordingly, if a function has a poor initial performance, its candidate would not be considered a promising one to be selected. To address this limitation, *No-Past-BO* (see A.4) method was introduced which uses a memory decay factor to discount the score over time (Vasconcelos et al., 2019). To further improve the performance, Vasconcelos et al. (2022) introduced *SETUP-BO* (see A.5), which auto-tunes the parameter while optimizing the objective function.

Since the initialization of the GP prior to the starting of optimization iterations plays a crucial role in the overall information that BO captures from the objective function, we modified the SETUP-BO implementation to incorporate random and grid initialization strategies. For random initialization, multiple samples drawn from a uniform distribution on the search space of the optimization problem

---

[1]The implementation of SETUP-BO was released https://github.com/thiago-vasconcelos/setup-bo by Vasconcelos et al. (2022)

Table 1: The table summarizes the Mann-Whitney U test results, which compared the performance of SETUP-BO with random initialization to other methods. A P-value $\leq 0.05$ indicates that the difference in performance is statistically significant, supporting the null hypothesis.

| Benchmark | Random Search | Grid Search | Deterministic Initialization | Grid Initialization |
|---|---|---|---|---|
| Hartmann 3D | $\approx 1.8 \times 10^{-48}$ | $\approx 3.8 \times 10^{-73}$ | $\approx 7.5 \times 10^{-165}$ | $\approx 1.1 \times 10^{-233}$ |
| Hartmann 6D | $\approx 2.6 \times 10^{-69}$ | $\approx 3.3 \times 10^{-154}$ | $\approx 1.7 \times 10^{-190}$ | $\approx 7.6 \times 10^{-76}$ |

are used to initialize the GP. For grid initialization, the search space is divided into a grid mesh, and the evaluation of the objective function on each corresponding point is used to initialize the GP.

## 3 EXPERIMENTS AND RESULTS

We compared five different methods including random search, grid search, SETUP-BO with deterministic initialization, random, and grid initialization on the Hartmann 6D and Hartmann 3D benchmark functions (see A.6), which are 6- and 3-dim uni-modal functions on a unit hyper-cube, respectively. The initialization population is equal for both random and grid initialization techniques.

Figure 1 demonstrates that SETUP-BO with random initialization outperforms other methods on the Hartmann 6D and Hartmann 3D benchmark functions by at least 6 and 93 times more minor errors, respectively. The convergence rate of SETUP-BO with random initialization is also better than other methods on these benchmarks. Notably, SETUP-BO with grid initialization performs poorly, potentially due to converging to local optima resulting from low uncertainty and inability to explore caused by uniform initialization in every region.

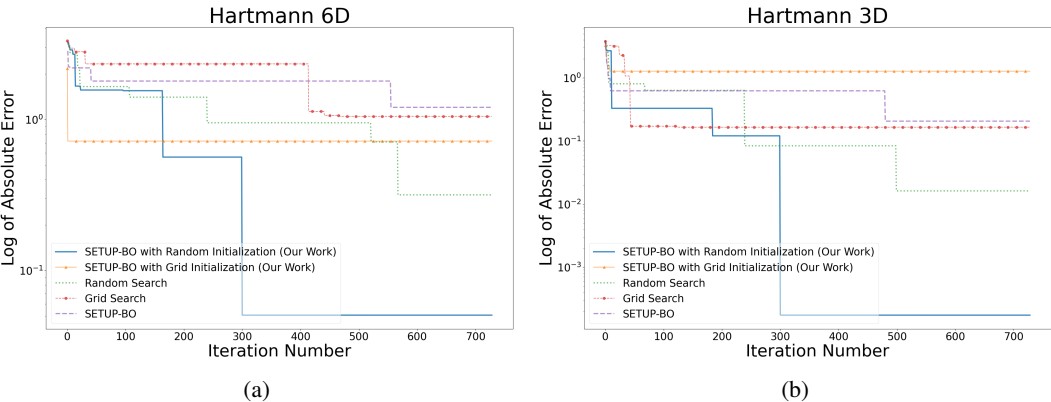

(a)                                                            (b)

Figure 1: Comparison of *log-absolute error* of the best solution found in each iteration w.r.t. the actual global optima by Random Search, Grid Search, SETUP-BO, SETUP-BO with Grid Initialization, and SETUP-BO with Random Initialization on Hartmann-6D (a) and Hartmann-3D (b)

We also performed a statistical test to compare the performance of SETUP-BO with random initialization against other methods. The Mann-Whitney U test which is a non-parametric test was used to compare two samples to test whether they were drawn from the same distribution. The test was implemented using the Scipy library, and the results are presented in table 1. Our null hypothesis was that the underlying distribution of the *absolute error* of SETUP-BO with random initialization was the same as that of the other methods, while our alternative hypothesis was that the distribution underlying SETUP-BO with random initialization was stochastically less than that of the other methods. Based on our analysis, we found evidence to support our alternative hypothesis, indicating that SETUP-BO with random initialization outperforms other methods in our study.

## 4 CONCLUSION

We investigated two simple initialization strategies for SETUP-BO: random and grid initialization. Our experiments on benchmark functions demonstrate that the random initialization strategy outperforms other methods. These findings suggest that this simple strategy can be effective in improving the performance of BO. The ideas of random and grid initialization strategies can also be applied to other population-based optimization methods.

ACKNOWLEDGEMENTS

We extend our sincere gratitude to Mr. Mohammad Azizmalayeri; his extensive experience and insightful feedback have helped us to assess our work from an unbiased perspective, and we are deeply grateful for his unwavering support. We are also grateful to the reviewers of our paper for their valuable feedback, which has significantly enhanced the quality of our research. Their constructive criticism has helped us to further improve our study and articulate our findings more effectively.

URM STATEMENT

In accordance with the guidelines set by the ICLR 2023 Tiny Papers Track, we affirm that Seyed Ali YaghoubNejad meets multiple URM criteria. As an author who belongs to an underrepresented group in machine learning, We recognize the importance of promoting diversity and inclusivity in our community. We are grateful for the opportunity to share our research and contribute to the advancement of the field.

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

## A APPENDIX

As mentioned Bayesian optimization has two major parts: the surrogate model and the acquisition function.

### A.1 SURROGATE MODEL IN BAYESIAN OPTIMIZATION

The surrogate model is responsible to capture the characteristics of the objective function and the uncertainty within. The surrogate model is modelled by a *Gaussian process*. Assume that $f : \mathbb{R}^D \to \mathbb{R}$ is the function we want to optimize and the evaluation on this function in $x_i$ is:

$$y_i = f(\boldsymbol{x}_i) + \epsilon_i \tag{1}$$

where $\epsilon_i \sim \mathcal{N}(0, \sigma^2)$ is a random noise. The Gaussian Process can be written as (Rasmussen, 2004):

$$p(\boldsymbol{f}|\boldsymbol{X}) = \mathcal{N}(\boldsymbol{f}|\boldsymbol{\mu}, \boldsymbol{K_f})$$
$$p(\boldsymbol{y}|\boldsymbol{f}) = \mathcal{N}(\boldsymbol{y}|\boldsymbol{f}, \sigma^2\boldsymbol{I})$$
$$p(\boldsymbol{y}|\boldsymbol{X}) = \mathcal{N}(\boldsymbol{y}|\boldsymbol{\mu}, \boldsymbol{K_f} + \sigma^2\boldsymbol{I}) \tag{2}$$

where $\boldsymbol{K}_f \in \mathbb{R}^{N \times N}$ is the covariance matrix calculated by the kernel function $k(.,.)$ (i.e $[\boldsymbol{K_f}]_{ij} = k(\boldsymbol{x}_i, \boldsymbol{x}_j)$).

In inference time, we are interested in predicting the value of function $f$ in $\boldsymbol{x}_*$ which indicates as $\boldsymbol{f}_*$:

$$\begin{bmatrix} \boldsymbol{f} \\ \boldsymbol{f}_* \end{bmatrix} = \mathcal{N} \left( \begin{bmatrix} \boldsymbol{\mu} \\ \boldsymbol{\mu}_* \end{bmatrix}, \begin{bmatrix} \boldsymbol{K} & \boldsymbol{K}_* \\ \boldsymbol{K}_*^T & \boldsymbol{K}_{**} \end{bmatrix} \right) \tag{3}$$

where $[\boldsymbol{\mu}]_i$ is the mean for $\boldsymbol{x}_i$. $\boldsymbol{K}$, $\boldsymbol{K}_*$ and $\boldsymbol{K}_{**}$ are the covariance matrix of train-train, train-test and test-test respectively.

### A.2 ACQUISITION FUNCTION IN BAYESIAN OPTIMIZATION

The main goal of the acquisition function is to suggest the next point to evaluate considering the captured data by the surrogate model (e.g. uncertainty of estimation in each point). There are 3 most used acquisition functions (Vasconcelos et al., 2019):

- *Probability of Improvement* that considers the probability for each point being better than the best found optimum (Kushner, 1964).

$$PI(\boldsymbol{x}) = p\{f(\boldsymbol{x}) \leq \boldsymbol{\mu}^- - \xi\} = \Phi \left( \frac{\boldsymbol{\mu}^- - \xi - \mu(\boldsymbol{x})}{\sigma(\boldsymbol{x})} \right) \tag{4}$$

  where $\boldsymbol{\mu}^- = min_i\mu(\boldsymbol{x}_i)$ and $\mu(\boldsymbol{x})$ and $\sigma(\boldsymbol{x})$ are the estimated mean and standard deviation of $\boldsymbol{x}$ respectively. $\Phi$ also indicates the CDF of standard Gaussian distribution. $\xi$ adjusts the balance between exploration and exploitation.

- *Expected improvement* that also take the amount of improvement into account (Mockus et al., 1978).

$$EI(\boldsymbol{x}) = \mathbb{E}[\boldsymbol{\mu}^- - \mu(\boldsymbol{x})] = \tau(\boldsymbol{x})\Phi \left( \frac{\tau(\boldsymbol{x})}{\sigma(\boldsymbol{x})} \right) + \sigma(\boldsymbol{x})\phi \left( \frac{\tau(\boldsymbol{x})}{\sigma(\boldsymbol{x})} \right) \tag{5}$$

  where $\tau(\boldsymbol{x}) = \boldsymbol{\mu}^- - \xi - \mu(\boldsymbol{x})$ and $\phi$ is the PDF of standard Gaussian distribution.

- *GP - Lower confidence bound* (or GP - Upper confidence bound in maximization problems) that use a linear combination of the mean and standard deviation of the GP (Brochu et al., 2010).

$$GP - LCB(\boldsymbol{x}) = \mu(\boldsymbol{x}) - k\sigma(\boldsymbol{x}) \tag{6}$$

  where $k$ adjust the balance between exploration and exploitation

Figure 2 and figure 3[2] show two consecutive iterations of BO with GP-UCB demonstrating how the surrogate model and acquisition function work to suggest the next candidate for evaluation.

---

[2]These figures were adapted from
https://github.com/fmfn/BayesianOptimization/blob/master/examples/visualization.ipynb.

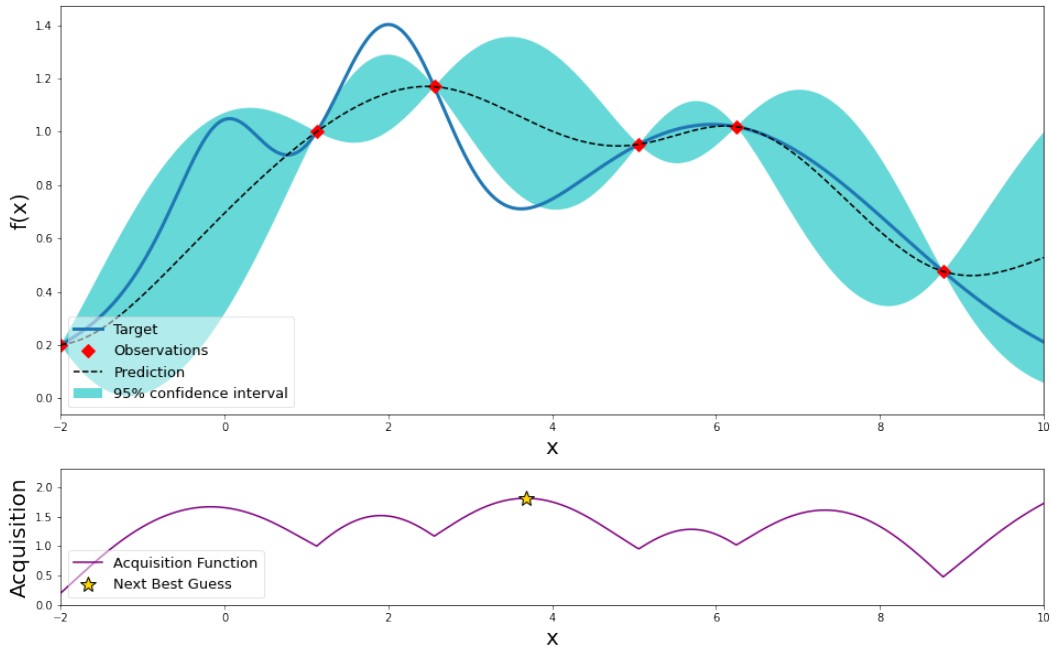

Figure 2: Top: The Bayesian Optimization in the $5^{th}$ iteration; the blue graph and the dotted black one show the objective function and the prediction of the objective function (a.k.a. surrogate model) respectively. The red dots indicate the observation, and the cyan region shows the uncertainty with a confidence level of 95%
Bottom: The acquisition function (GP-UCB) and the star shows the next candidate for evaluation

### A.3   GP-HEDGE ALGORITHM

The *GP-Hedge algorithm* (Auer et al., 1995) examines a predetermined collection of acquisition functions, organized as a portfolio. Its fundamental principle revolves around the nomination of a candidate point for evaluation by all acquisition functions during each iteration of Bayesian Optimization. These acquisition functions, utilizing the same surrogate model, collaboratively contribute to the selection process outlined in Algorithm 1. The probability of choosing a nominated point is contingent upon the past performance of the associated acquisition function.

---

**Algorithm 1** GP-Hedge Algorithm (Vasconcelos et al., 2022)

---

Select hyperparameter $\eta \in \mathbb{R}^+$
Set $G_j^0 = 0$ for $j = 1, 2, ..., J$ $\qquad\qquad\triangleright$ $G_j^t$ is the total score associated with the acquisition function $j$ up to the time $t$
**for** $t = 1, 2, ...$ **do**
$\quad$ Nominate points from each acquisition function $h_j$: $\boldsymbol{x}_j^t = \arg\max_{\boldsymbol{x}} h_j(\boldsymbol{x})$
$\quad$ Select a nominee $\boldsymbol{x}^t = \boldsymbol{x}_j^t$ with probability $p_j^t = \dfrac{\exp\left(\eta G_j^{t-1}\right)}{\sum_{j'=1}^{J} \exp\left(\eta G_{j'}^{t-1}\right)}$
$\quad$ Compute $y^t$ by evaluating the objective on point $\boldsymbol{x}^t$
$\quad$ Augment the data $\mathbb{D}^t$ with the new pair $\left(\boldsymbol{x}^t, y^t\right)$ $\qquad\triangleright$ $\mathbb{D}^t$ is the set of observations up to the time $t$
$\quad$ Update the surrogate GP model
$\quad$ Update the rewards $G_j^t = G_j^{t-1} - \mu\left(\boldsymbol{x}_j^t\right)$ from the updated GP posterior
**end for**

---

It should be noted that in the GP-Hedge algorithm, the variable $G_j^t$ represents the cumulative sum of the scores $score_j(\boldsymbol{x}_j^{t'})$ from $t = 1$ to $t$. In the GP-Hedge algorithm, the score $score_j(\boldsymbol{x}_j^t)$ corre-

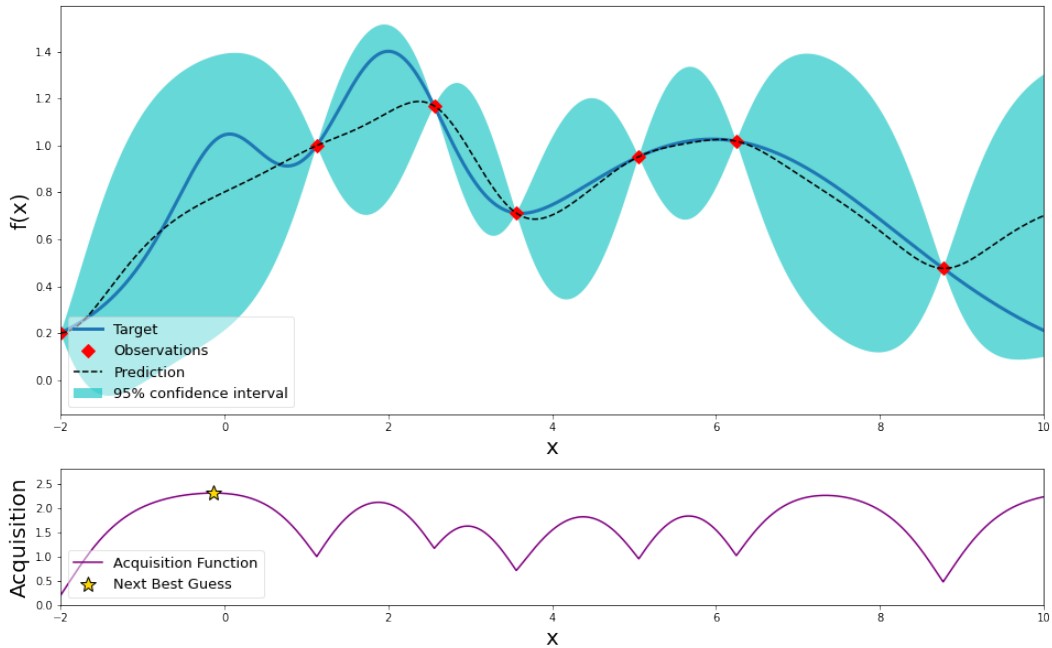

Figure 3: Top: The Bayesian Optimization in the $6^{th}$ iteration; As demonstrated, the uncertainty around the evaluated point is decreased.
Bottom: The acquisition function (GP-UCB) and the star shows the next candidate for evaluation.

sponds to the evaluation $\mu_j(\boldsymbol{x}_j^t)$ of the surrogate model on the point $\boldsymbol{x}_j^t$ suggested by the acquisition function $j$ after updating the surrogate model with the dataset $\mathbb{D}^t$. It is important to mention that the original GP-Hedge algorithm was initially designed for solving maximization problems. However, in the case of minimization problems, a modification is introduced where $G_j^t$ is computed as

$$G_j^t = -\sum_{t'=1}^{t} \mu_j(\boldsymbol{x}_j^{t'}) \tag{7}$$

### A.4   NO-PAST-BO ALGORITHM

The *No-PASt-BO* algorithm (Vasconcelos et al., 2019) introduces two modifications to the GP-Hedge algorithm. It incorporates a memory factor, denoted as $0 \le m \le 1$, during the reward update process to reduce the influence of past evaluations as the optimization progresses.

$$G_j^t = mG_j^{t-1} - \mu\left(\boldsymbol{x}_j^t\right) \tag{8}$$

By utilizing the memory factor, the algorithm assigns less weight to previous rewards. However, this adjustment can result in similar rewards at later iterations, causing the algorithm to select acquisition functions randomly from the portfolio. To address this issue, the No-PASt-BO algorithm introduces a normalization step to ensure proper selection and maintain the effectiveness of the acquisition functions.

$$r_j^t = \frac{G_j^t - r_{\max}^t}{r_{\min}^t - r_{\max}^t} \tag{9}$$

Here $r_{\max}^t = \max_j G_j^t$, and $r_{\min}^t = \min_j G_j^t$. The full algorithm is presented in Algorithm 2

### A.5   SETUP-BO ALGORITHM

In the BO context, the usage of hyperparameters introduces challenges. The evaluation of the costly objective function makes it undesirable (and often infeasible) to test multiple candidate values for

---

**Algorithm 2** No-Past-BO Algorithm (Vasconcelos et al., 2022)

---

Select hyperparameter $\eta \in \mathbb{R}^+$
Select hyperparameter $m \in [0, 1]$
Set $G_j^0 = 0$ for $j = 1, 2, ..., J$
**for** $t = 1, 2, ...$ **do**
    Nominate points from each acquisition function $h_j$: $\boldsymbol{x}_j^t = \arg\max_{\boldsymbol{x}} h_j(\boldsymbol{x})$
    Compute $r_{\max}^t = \max\limits_{j} G_j^t$
    Compute $r_{\min}^t = \min\limits_{j} G_j^t$
    Compute the normalized rewards: $r_j^t = \dfrac{G_j^t - r_{\max}^t}{r_{\min}^t - r_{\max}^t}$
    Select a nominee $\boldsymbol{x}^t = \boldsymbol{x}_j^t$ with probability $p_j^t = \dfrac{\exp\left(\eta r_j^{t-1}\right)}{\sum_{j'=1}^{J} \exp\left(\eta r_{j'}^{t-1}\right)}$
    Compute $y^t$ by evaluating the objective on point $\boldsymbol{x}^t$
    Augment the data $\mathbb{D}^t$ with the new pair $\left(\boldsymbol{x}^t, y^t\right)$
    Update the surrogate GP model
    Update the rewards $G_j^t = mG_j^{t-1} - \mu\left(\boldsymbol{x}_j^t\right)$ from the updated GP posterior
**end for**

---

finding the best set of hyperparameters. Thus, it is crucial to develop a strategy to obtain the best hyperparameters for each optimization task. To address this problem, the *SETUP-BO Algorithm* (Vasconcelos et al., 2022) based on the use of *Thompson Sampling* (TS) (Russo et al., 2018) was proposed, which allows the automatic adaptation of hyperparameters in each iteration. The portfolio of acquisition functions can be viewed as a *multi-armed bandit* problem, where the selection of an acquisition function that leads to the best evaluation of the black-box function is necessary. Therefore, applying TS to assist in the selection of the acquisition function is considered a promising approach.

Two hyperparameters introduced in the No-PASt-BO algorithm (Vasconcelos et al., 2019), namely the memory factor ($m$) and the $\eta$ hyperparameter, are handled by the TS-based approach. For the memory factor parameter $m$, a Beta probability distribution was chosen as the prior, since it covers the range of possible values for $m$ in the interval $[0, 1]$. A Bernoulli likelihood function is chosen, where a success (a value equal to one) is defined by a result better than the best result achieved so far. Conversely, a failure (a zero value) is considered. Consequently, obtaining the posterior distribution becomes easy, as Beta and Bernoulli are conjugate distributions.

Regarding the $\eta$ value, a Gamma probability distribution was selected as the prior due to its positive domain ($\eta \in \mathbb{R}^+$). Another reason for choosing the Gamma distribution is its conjugacy with the exponential distribution. However, the likelihood function takes the form of the Boltzmann distribution, representing the probability of selecting a certain acquisition function given the energy of the score. The Boltzmann likelihood is not conjugate with the Gamma prior, which means that a closed-form solution for the corresponding posterior does not exist. Therefore, approximate inference is required.

To achieve analytical conjugate posterior updates, some additional considerations are made. It is a fact that if all the rewards are not equal, there will be at least one $r_{\max=0}$ and one $r_{\min=-1}$, due to the normalization step. Considering a portfolio of three acquisition functions, the probability of choosing an acquisition function $j$ is approximated as:

$$p_j^t = \frac{\exp\left(\eta r_j^{t-1}\right)}{1 + \exp\left(-\eta\right) + \exp\left(\eta r_{\text{intermediary}}^{t-1}\right)} \approx q_j^t = C \exp\left(\eta r_j^{t-1}\right) \tag{10}$$

where $C$ is the constant and represents the inverse of the mean value of the denominator. Although this approximation is not entirely accurate, it respects the fact that higher values of $\eta$ should be associated with small values of $r_j^{t-1}$ and low values otherwise. Importantly, this approximation transforms the problem into a Gamma-exponential conjugate pair, for which a closed-form expression for the posterior distribution exists. The final algorithm is summarized in Algorithm 3.

---

**Algorithm 3** SETUP-BO Algorithm (Vasconcelos et al., 2022)

---

Set $G_j^0 = 0$ for $j = 1, 2, ..., J$
**for** $t = 1, 2, ...$ **do**
    TS step: Sample hyperparameter $\eta \sim Gamma(\alpha, \beta)$
    TS step: Sample hyperparameter $m \sim Beta(a, b)$
    Nominate points from each acquisition function $h_j$: $\boldsymbol{x}_j^t = \arg\max_{\boldsymbol{x}} h_j(\boldsymbol{x})$
    Compute $r_{\max}^t = \max\limits_j G_j^t$
    Compute $r_{\min}^t = \min\limits_j G_j^t$
    Compute the normalized rewards: $r_j^t = \dfrac{G_j^t - r_{\max}^t}{r_{\min}^t - r_{\max}^t}$
    Select a nominee $\boldsymbol{x}^t = \boldsymbol{x}_j^t$ with probability $p_j^t = \dfrac{\exp\left(\eta r_j^{t-1}\right)}{\sum_{j'=1}^J \exp\left(\eta r_{j'}^{t-1}\right)}$
    Compute $y^t$ by evaluating the objective on point $\boldsymbol{x}^t$
    Augment the data $\mathbb{D}^t$ with the new pair $\left(\boldsymbol{x}^t, y^t\right)$
    Update the surrogate GP model
    Update the rewards $G_j^t = m G_j^{t-1} - \mu\left(\boldsymbol{x}_j^t\right)$ from the updated GP posterior
    **if** $y^t$ is the best point evaluate so far **then**
        Update the posteriors $a = a + 1$
    **else**
        Update the posteriors $b = b + 1$
    **end if**
    Update the posteriors $\alpha = \alpha + 1$
    Update the posteriors $\beta = \beta + r_j^t$
**end for**

---

## A.6 BENCHMARK FUNCTIONS

In this section, a comprehensive description of the benchmark objective functions employed in our study is presented. The functions under consideration include the Hartmann 3D function and the Hartmann 6D function, where the results of both benchmarks are depicted in figure 1. Furthermore, we analyze the Branin function, illustrated in figure 6. To evaluate and compare the performance of these functions, we conducted the Mann-Whitney U test, and the whole results are summarized in table 2.

1. Hartmann 6D: The Hartmann 6D[3] function is a widely used benchmark function for testing global optimization algorithms. It is a six-dimensional function that is complex and multi-modal, making it challenging to optimize. Its mathematical formula is given by:

$$f(\boldsymbol{x}) = -\sum_{i=1}^{4} \alpha_i \exp\left(-\sum_{j=1}^{6} A_{ij}(x_j - P_{ij})^2\right) \tag{11}$$

where $\boldsymbol{x} = (x_1, \ldots, x_6)$ is the input vector, and the parameters $\alpha_i$, $A_{ij}$, and $P_{ij}$ are pre-defined constants. The global minimum value is approximately -3.32237, and the optimal point is given by the vector:

$$\boldsymbol{x}^* = (0.20169, 0.150011, 0.476874, 0.275332, 0.311652, 0.6573) \tag{12}$$

The detailed result is depicted in figure 4

2. Hartmann 3D: The Hartmann 3D[4] function is a three-dimensional version of the Hartmann function, commonly used to test global optimization algorithms. It is similar to the six-

---

[3]https://www.sfu.ca/ ssurjano/hart6.html
[4]https://www.sfu.ca/ ssurjano/hart3.html

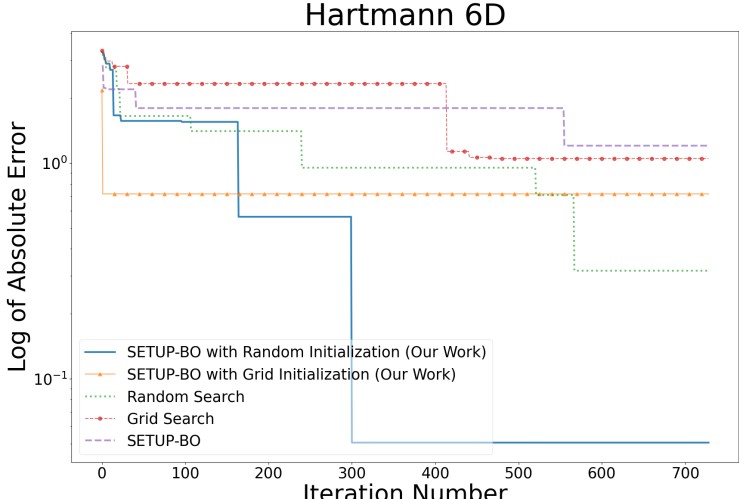

Figure 4: Comparison of *log-absolute error* of the best solution found in each iteration w.r.t. the actual global optima by Random Search, Grid Search, SETUP-BO, SETUP-BO with Grid Initialization, and SETUP-BO with Random Initialization on Hartmann 6D

dimensional version but has a simpler input space. Its mathematical formula is given by:

$$f(\boldsymbol{x}) = -\sum_{i=1}^{4} \alpha_i \exp\left(-\sum_{j=1}^{3} A_{ij}(x_j - P_{ij})^2\right) \tag{13}$$

where $\boldsymbol{x} = (x_1, x_2, x_3)$ is the input vector, and the parameters $\alpha_i$, $A_{ij}$, and $P_{ij}$ are predefined constants. The global minimum value is approximately -3.86278, and the optimal point is given by the vector:

$$\boldsymbol{x}^* = (0.114614, 0.555649, 0.852547) \tag{14}$$

The detailed result is illustrated in figure 5

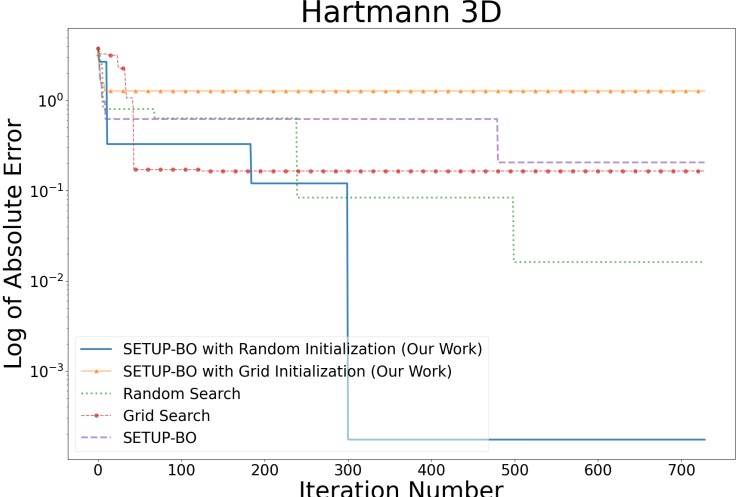

Figure 5: Comparison of *log-absolute error* of the best solution found in each iteration w.r.t. the actual global optima by Random Search, Grid Search, SETUP-BO, SETUP-BO with Grid Initialization, and SETUP-BO with Random Initialization on Hartmann 3D

Table 2: This table summarizes the Mann-Whitney U test results, which compared the performance of SETUP-BO with random initialization to other methods on the Hartmann 3D, Hartmann 6D, and Branin. A P-value $\leq 0.05$ indicates that the difference in performance is statistically significant, supporting the null hypothesis.

| | Benchmark | Random Search | Grid Search | Deterministic Initialization | Random Initialization | Grid Initialization |
|---|---|---|---|---|---|---|
| **Random BO** | Hartmann 3D | $1.8 \times 10^{-48}$ | $3.8 \times 10^{-73}$ | $7.5 \times 10^{-165}$ | 0.5 | $1.1 \times 10^{-233}$ |
| | Hartmann 6D | $2.6 \times 10^{-69}$ | $3.3 \times 10^{-154}$ | $1.7 \times 10^{-190}$ | 0.5 | $7.6 \times 10^{-76}$ |
| | Branin | $5.5 \times 10^{-177}$ | $8.2 \times 10^{-170}$ | $1.1 \times 10^{-204}$ | 0.5 | $1.3 \times 10^{-167}$ |
| **Grid BO** | Hartmann 3D | 1 | 1 | 1 | 1 | 0.5 |
| | Hartmann 6D | $2.2 \times 10^{-52}$ | $1.2 \times 10^{-280}$ | $4.2 \times 10^{-287}$ | 1 | 0.5 |
| | Branin | $4.7 \times 10^{-69}$ | 1 | $1 \times 10^{-201}$ | 1 | 0.5 |

3. Branin: The Branin function[5] is a two-dimensional function with three global minima and many local minima, often used to test optimization algorithms' ability to escape local optima. The result for the Branin is shown in figure 6. Its mathematical formula is given by:

$$f(\boldsymbol{x}) = a(x_2 - bx_1^2 + cx_1 - r)^2 + s(1 - t)\cos(x_1) + s \tag{15}$$

where $a$, $b$, $c$, $r$, $s$, and $t$ are pre-defined constants. The global minimum value is approximately 0.397887, and there are three global optimal points, which are:

$$
\begin{aligned}
\boldsymbol{x}_1^* &= (-\pi, 12.275) \\
\boldsymbol{x}_2^* &= (\pi, 2.275) \\
\boldsymbol{x}_3^* &= (9.42478, 2.475)
\end{aligned}
\tag{16}
$$

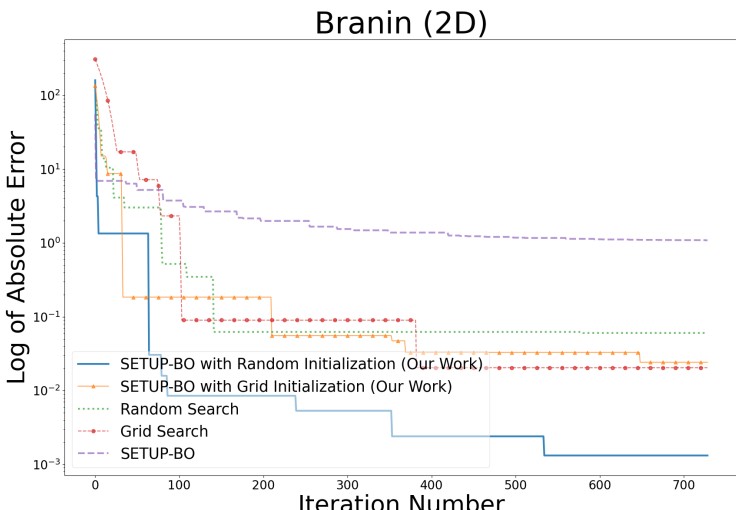

Figure 6: Comparison of *log-absolute error* of the best solution found in each iteration w.r.t. the actual global optima by Random Search, Grid Search, SETUP-BO, SETUP-BO with Grid Initialization, and SETUP-BO with Random Initialization on Branin

---

[5]https://www.sfu.ca/ ssurjano/branin.html

