# OpenReview forum: "Exploring Efficient and Simple Initialization Strategies for Bayesian Optimization with SETUP-BO"
_ICLR.cc/2023/TinyPapers — Submitted to Tiny Papers @ ICLR 2023_

### Official Review · Reviewer_6xHM · 2023-03-29

**Confidence:** 4

**Summary Of Contributions:**

This study investigates the usefulness of random and grid initialization strategies for a self-tuned Bayesian optimization approach. Author(s) tested their recommended initialization techniques against deterministic initialization to see how they performed. According to their results, random initialization can enhance Bayesian optimization performance.

**Rating:**

High Potential (HP): a submission which meets the reviewing criteria and has potential to make an impact on the field

**Strengths And Weaknesses:**

## Strengths
- The article demonstrate clarity and simplicity in their work.
- The work presented is reproducible and therefore easy to compare their work with other existing SOTA.
- The submission follows the basic requirements of ICLR code of conduct.
- The findings demonstrated in the article are justified based on the work shown in the appendix section and source code link provided.

## Weakness
- The figure quality is not very good and therefore text are hardly visible
- The benchmarks shown in the article are not enough for the kind of work demonstrated by the author.

**Suggested Changes:**

- The better quality of figures and tables would be good.
- More benchmarks should be compared with to demonstrated the results achieved by your setup.
- Minor grammatical mistakes can be seen in the test. Correcting them would make test more effective.

---

### Official Review · Reviewer_3eJE · 2023-03-31

**Confidence:** 4

**Summary Of Contributions:**

This paper investigates the efficacy of random initialization and grid initialization strategies  in comparison to traditional initialization strategies. The authors find that random initialization strategies outperform deterministic initialization for the SETUP-BO algorithm.

**Rating:**

Clear, Correct, and Reproducible (CCR): a submission which meets the reviewing criteria

**Strengths And Weaknesses:**

Strengths
-  The paper provides strong empirical results that validate the hypothesis that poor initialization can result in inefficient optimization. Based on the results provided, the randomization strategy does outperform deterministic strategies.
- The experiments are well-designed and can be reproducible based on the provided description,

Weakness
- The formatting and language of the paper require significant editing.
- Random initialization is not a novel idea. However, the paper does provide empirical results to support the methodology.

**Suggested Changes:**

- Consider using a more formal tone when writing the paper. For example, the Abstract mentions a "bunch of experiments" which is underspecified.
- The paper should go through another editing pass, especially around the descriptions of the figures.

---

### Official Review · Reviewer_8jhu · 2023-04-03

**Confidence:** 2

**Summary Of Contributions:**

This paper studies the effect of different approaches for selecting initial evaluation points of function in the context of Bayesian optimisation. They observe that in the settings they study random selection of initial sampling outperforms other deterministic approaches.

**Rating:**

Needs Clarification (NC): a submission which does not meet the reviewing criteria and needs clarification for its described problem or solution

**Strengths And Weaknesses:**

++


+ Bayesian optimisation is widely used in several applied sciences. Thus methods that optimize its performance are expected to be influential.



--

- There were some parts in the text that were unclear and require additional explanation. In particular the previous approach the authors extend needs some brief description, as well as the extensions they propose.

**Suggested Changes:**

- Please give the definition of the Hartmann functions you used in your experiments.


- I would suggest to increase the font size in the plots.

- Please explain briefly in the appendix how the method SETUP-BO works. The paper that you cite is non-open access and your method relies heavily on their method.

- Please clarify what is "Mann-Whitney U test P-value of SETUP-BO".

- Table 1 needs clarification, and in particular what the entries are representing. Please clarify because it seems that I am missing the main point here. If they are p-values it seems to me that all different variants deliver extremely small p-values.

- Can you give some intuitive explanation on why the random sampling performed better than the grid sampling?

---

### Author Response · Authors · 2023-05-30
**Summary of Revised Changes in Response to Reviewers' Feedback and Decision to Opt-in for Archival**

We have revised our paper for ICLR 2023, Tiny Papers, based on the invaluable feedback provided by the reviewers. Here is a summarized list of changes:

1. **Adding detailed explanation about previous works:** In response to the request of one of the reviewers, we have included a comprehensive explanation of the previous works in the appendix section. This addition provides readers with a thorough understanding of the background and context of our research.

2. **Complete specification of benchmark functions:** To increase clarity, we have included a complete specification of the benchmark functions, namely Hartmann 6D, Hartmann 3D, and Branin, in the appendix section.

3. **Increasing the font size in plots:** With the aim of improving readability, we have increased the font size in the plots. Furthermore, larger plots of our tests have been added in the appendix section.

4. **Discussing the Mann-Whitney U test and p-values:** We have addressed the reviewers' concerns by providing a detailed discussion about the Mann-Whitney U test, including our null and alternative hypotheses, and the implementation of the test. Additionally, we have attached a comprehensive table of results to the appendix. In response to a reviewer's inquiry regarding the small p-value, we attribute it to the significant number of iterations in which the SETUP-BO with random initialization algorithm reported superior solutions compared to its counterparts.

5. **Intuitive comparison of random sampling versus grid sampling:** We can provide an intuitive explanation for why random sampling performed better than grid sampling in our study. The main reason lies in the inherent limitation of grid search in finding the global optimum when it lies between grid points. Grid search relies on evaluating predefined points on a grid, which may not capture the true location of the global optimum if it falls in between those points. On the other hand, random sampling does not suffer from this limitation as it explores the search space in a more flexible and unrestricted manner. Furthermore, we have proposed insights about the potential impact of uniform grid initialization in different regions on the performance of SETUP-BO. However, we acknowledge that further investigation is necessary to validate this assumption.

6. **Enhancing of writing style:** Taking into account the reviewers' comments, we have improved the level of formality in specific sections of the paper without compromising the contextual meaning.

7. **Adding an extra benchmark and reference:** Following reviewer suggestions, we have incorporated an additional benchmark into our experiments. However, due to the constraint of the 2-page limit, we were not able to include further experiments. Additionally, we have included the [1] reference as it was recommended.

8. **Acknowledgement:** We have included an acknowledgement section to express our heartfelt gratitude to the reviewers and other colleagues for their thoughtful comments.

In conclusion, **we express our desire to opt-in for archival.**

---

### Meta-Review · Area_Chair_KjTQ · 2023-04-02

**Recommendation:** Invite to present
**Confidence:** 4

**Metareview:**

Strengths:
- A key strength of the work is the well-designed and reproducible experiments.
- The paper has a well-focused and interesting goal of comparing different initialization strategies for Bayesian Optimization w.r.t. their efficiency.
- The findings are well-organized and clear evidence is provided.

Areas to improve:
- The main limitation is there is scope for improving the writing, and a careful proofread and attempt to improve the presentation along reviewer suggestions is recommended.
- More experiments/benchmarks would be useful to further substantiate the claims made in the paper.



**Summary:**

The authors investigate various initialization strategies in comparison to traditional initialization strategies for the SETUP-BO algorithm, and find that random initialization strategies perform well. The experiments are well-designed and reproducible but could be made more extensive.

**Comments And Feedback To The Authors:**

Please incorporate writing related feedback and suggestions from the reviewers to improve the paper presentation.

Random initialzation is not a new technique and therefore a reference to the literature should be provided, for example [1] below.
- [1] Bergstra, James, and Yoshua Bengio. "Random search for hyper-parameter optimization." Journal of machine learning research 13, no. 2 (2012).

Comparison for more benchmarks (e.g. more rows in Table 1) would help substantiate the claims/observations further.

**Reason For Not Giving A Higher Recommendation:**

The problem studied and experiments designed to study are interesting and useful for the community, but the experiments should be more thorough to substantiate the claims (only two benchmarks are studied which are related).

**Reason For Not Giving A Lower Recommendation:**

Both reviewers agree the paper satisfies CCR, and the meta-reviewer agrees.

---

### Decision · Program_Chairs · 2023-04-07

Invite to present